# The Virtual “Enfacement Illusion” on Pain Perception in Patients Suffering from Chronic Migraine: A Study Protocol for a Randomized Controlled Trial

**DOI:** 10.3390/jcm11226876

**Published:** 2022-11-21

**Authors:** Sara Bottiroli, Marta Matamala-Gomez, Marta Allena, Elena Guaschino, Natascia Ghiotto, Roberto De Icco, Grazia Sances, Cristina Tassorelli

**Affiliations:** 1Faculty of Law, Giustino Fortunato University, 82100 Benevento, Italy; 2Headache Science and Neurorehabilitation Center, IRCCS Mondino Foundation, 27100 Pavia, Italy; 3Mind and Behavior Technological Center, Department of Psychology, University of Milano-Bicocca, 20126 Milan, Italy; 4Department of Brain and Behavioral Sciences, University of Pavia, 27100 Pavia, Italy

**Keywords:** pain, chronic migraine, virtual reality, embodiment, enfacement illusion, body image

## Abstract

Background: given the limited efficacy, tolerability, and accessibility of pharmacological treatments for chronic migraine (CM), new complementary strategies have gained increasing attention. Body ownership illusions have been proposed as a non-pharmacological strategy for pain relief. Here, we illustrate the protocol for evaluating the efficacy in decreasing pain perception of the enfacement illusion of a happy face observed through an immersive virtual reality (VR) system in CM. Method: the study is a double-blind randomized controlled trial with two arms, involving 100 female CM patients assigned to the experimental group or the control group. The experimental group will be exposed to the enfacement illusion, whereas the control group will be exposed to a pleasant immersive virtual environment. Both arms of the trial will consist in three VR sessions (20 min each). At the baseline and at the end of the intervention, the patients will fill in questionnaires based on behavioral measures related to their emotional and psychological state and their body satisfaction. Before and after each VR session, the level of pain, the body image perception, and the affective state will be assessed. Discussion: this study will provide knowledge regarding the relationship between internal body representation and pain perception, supporting the effectiveness of the enfacement illusion as a cognitive behavioral intervention in CM.

## 1. Introduction

Migraine is known as one of the most prevalent and disabling neurological disorders worldwide [1]. Usually, it recurs in an episodic pattern (less than 15 days/month—episodic migraine), but in a small, though clinically relevant portion of sufferers, migraine may acquire an aggressive evolution being present on 15 or more days/month. This is so-called chronic migraine (CM) [2]. CM is mainly managed through pharmacological treatment, requiring both the acute drugs taken during attacks and the preventive treatment aimed at reducing the frequency of attacks [3]. Unfortunately, available treatment options have limited efficacy (50% reduction of migraine days in 50% of patients), they may be poorly tolerated, and they can induce clinically relevant side effects [4,5]. In addition, when considering that CM is a complex neurological disorder that may manifest with variable phenotypical profiles and response to treatments, it is important to tailor the treatment approach to patient’s characteristics and inter-individual differences. In this regard, it should be noted that CM is often associated with several comorbidities [6], including those of a psychological nature, such as dependence behaviors, anxiety, depression, and personality disorders [7,8,9,10,11,12], which could affect the course of treatment [7,8]. As consequence, persistent pain associated to CM poses a high socio-economic burden [13,14], including the costs of pharmacological treatments, primary care visits, and hospitalizations, together with the impact on the patient’s productivity and everyday life [14,15]. Because of this, recent years have seen a proliferation of new therapeutic strategies to address migraine-related pain, as well as new techniques that offer promising approaches for development that could be integrated into pharmacological options.

Nowadays there is an increasing interest in the integration of new technologies for clinical practices [16]. In particular, virtual reality (VR) has been widely used to promote mental health in populations presenting different clinical conditions [17]. Recently, several investigations showed the effectiveness of a VR-based approach for pain relief in patients suffering from chronic pain [18,19,20]. Indeed, VR has been shown as being effective as a treatment strategy for burn pain, acute pain, or induced pain [21,22,23,24,25,26]. VR as an analgesic option that can be used in two modalities: as a distraction therapy through immersiveness, and as a tool for modulating body representation through virtual body ownership illusions (BOI) [18,24,27,28]. Distraction therapy consists in temporally diverting the patient’s attention from the pain through the VR experience [29]. In detail, the cognitive mechanisms that mediate VR-based interventions as a distraction therapy for pain relief are attention, concentration, and emotional alteration [30]. Moreover, the immersiveness on such VR environments allows participants to fully interact with the environment increasing the distraction from pain [28,31,32].

The use of virtual body ownership illusions for pain relief relies on inducing the sense of embodiment toward virtual bodies for changing body representation [33,34]. The sense of “embodiment” refers to the sense of having a body [35] and it is the result of a complex interaction between bottom-up and top-down sensory information [36]. According to this, the sense of embodiment consists in three subcomponents: the sense of self-location, the sense of agency, and the sense of body ownership [33]. Once embodiment is successfully induced, the subject feels as though they are inside a body that is moving according to his/her own intentions and interacting with the environment [33]. In the last few years, there has been a proliferation of studies trying to understand how to experimentally manipulate bodily perception and embodiment by using virtual body ownership illusions [37,38]. This is possible because a main feature of the VR system is its capability to induce a sense of ‘presence’, that is the sense of ‘being there’ within the virtual environment [39,40,41]. According to this, a large number of studies have tested virtual body ownership illusions by using synchronous visuo-tactile or visuo-motor correlations for inducing the sense of embodiment in a VR environment where participants could feel fully immersed into and present in the generated virtual world. [42,43,44,45]. In this regard, it has been demonstrated that watching the face of another person while that face and one’s own face are stroked synchronously induces the illusion of self-recognition toward the other face, the so-called “enfacement illusion” [46,47].

The enfacement illusion is the result of the plasticity of the self-face representation, which can be temporarily modified to include another person’s facial features [46,47]. Hence, the enfacement illusion has resulted as a good strategy for changing self-representation, with important implications for all those subjects who have distorted body representations, including patients suffering from chronic pain [48,49,50]. Indeed, a recent study showed that it is possible to induce the enfacement illusion by using VR systems [51]. The enfacement illusion is associated with improved emotional contagion and emotional recognition. It has indeed demonstrated that there is a sort of “mood migration” from the virtual face to another person’s face in case of synchronicity between them [52]. In line with this point, a recent study [53] has shown that by applying synchronous visuo–tactile stimulation observed through videos, it is possible to boost facial mimicry, which is the automatic imitation of another person’s emotion. From a neuroimaging point of view, the virtual experience derived from the enfacement illusion is able to induce changes in the sensory–motor cortex [54], and in particular in those areas that are the neural basis of body image [55] and emotion processing [56,57,58], that is, the primary sensory (S1) and motor (M1) cortices. Interestingly, these brain areas are involved in the cerebral dimensions of pain: one representing the discriminative dimension (S1) and one representing the affective dimension (M1).

It has been indeed shown that the perception of pain can alter both facial recognition and visuospatial perception in CM compared to healthy subjects [59]. Furthermore, it has been found that by reducing the altered perception of body image in CM patients with medication overuse, it is possible to induce beneficial effects on their affective state and on their perception of pain [60]. Moreover, there is evidence that electrophysiological homeostasis is altered abnormally (allostatic load) in a migraine brain with regard to what concerns sensory–motor network interactions [61,62,63,64,65] as well as neurostimulation techniques applied at the right M1 or S1 which can effectively decrease migraine pain frequency, duration, and intensity [66]. Recently our research group has evaluated the effects of an intervention based on visual feedback—i.e., intended to modify pain through the perception of the body image by means of the observation of facial expressions with different emotional content [67]—on the modulation of pain perception in a sample of female CM patients [68]. Interestingly, we showed that the simple observation of a facial expression with positive emotional content, when compared to other emotional content (negative or neutral), was able to decrease pain perception. The results from this study have paved the way for the integration of new technologies, such as VR systems, for modulating body representation through the use of virtual body ownership illusions, in particular through the use of the ‘enfacement illusion’. To the best of our knowledge, no study has used the enfacement illusion for the treatment of pain in patients suffering from CM.

The present study protocol presents a longitudinal randomized controlled trial (RCT) protocol, which aims to investigate whether it is possible to decrease pain perception in patients with CM by inducing the enfacement illusion of representing oneself in a happy face through a VR system (experimental group). It is expected that this effect is mediated by an improved body image perception and empathy for positive emotions. Our previous findings [68] allowed us to explore the different effects of being exposed to neutral or sad stimuli. Hence, in the present study, we used the positive emotional face as being the only one recognized as effective for pain relief in patients with CM. According to this, the main aim of this study is to enucleate specifically the effect of enfacement from the possible synergic effect of the exposure to a positive stimulus (happy face). In the control group, the patients will be subjected to a positive exposure (pleasant environment) in an immersive VR environment, which has already been demonstrated to be able to produce distracting effects on pain perception in patients with CM [69]. The rationale for choosing this control condition derives from the willingness to use another positive visual stimulus in immersive VR, which is already known to be effective in distracting individuals from pain in CM [69], that can serve to appreciate the additional effects of producing the enfacement illusion of a happy face in the experimental group. It is hypothesized that patients in the experimental group will experience greater pain relief and will improve the perception of their body image compared to the control group. Therefore, the present protocol study aims to show a possible relationship between the manipulation of the body image and pain perception and to demonstrate the effectiveness of the use of the virtual enfacement illusion as a cognitive behavioral intervention for pain relief in CM.

## 2. Materials and Methods

### 2.1. Study Design

The proposed study is a prospective double-blind RCT with two arms. A CONSORT flow chart for enrollment and randomization is shown in Figure 1. Patients fulfilling inclusion criteria, once they have signed the informed consent form, will undergo a baseline assessment (T0) using the below-listed tests and will be randomized to one of two groups: experimental and control. Both groups will be exposed to three VR sessions each lasting approximately 20 min during a one-week period. The first session will be carried out immediately after enrollment and T0, whereas the next two will be carried out in the following days when the patients will report to have a migraine attack. At T0 and at the end of the intervention (T1), the patients will fill in a questionnaire based on behavioral measures related to their emotional and psychological state and their body image perception. T1 assessment will then be carried out at the end of the last VR session. Before and after each VR session the level of pain, body image, and affective state of the patients will be assessed. Furthermore, after each VR session, the sense of embodiment (experimental group) or the sense of immersiveness (control group) will be assessed.

This study protocol conforms to the Standard Protocol Items: Recommendations for Interventional Trials (SPIRIT) guidelines (see Appendix A).

### 2.2. Study Setting

This trial will take place at the Headache Science and Neurorehabilitation Center of the IRCCS National Neurological Institute Mondino Foundation (Pavia), a tertiary referral center for the diagnosis and care of migraine in Northern Italy.

### 2.3. Participant Recruitment and Eligibility Criteria

Participants will be recruited among patients referred to the Center. An expert neurologist will verify the eligibility criteria during the recruitment process based on the participants’ history, headache diaries, and neurological evaluation.

Inclusion criteria for patients will be: (a) 18–65 years of age; (b) female sex; (c) fulfilment of the ICHD-3 criteria [2] for CM (with or without medication overuse); (d) previous history of migraine as primary headache; (e) an intensity of migraine attacks between 20 and 80 on a 0–100 VAS. Only females will be included in this study because of the reported differences on pain perception between males and females [70].

Exclusion criteria will be: (a) epilepsy, psychosis, intellectual disability, pregnant women and breastfeeding women; (b) visual problems; (c) presence of chronic non-cephalic pain.

All patients will receive the advice to avoid any abortive medication during the study period. Preventive treatments will be allowed given that they are not expected to affect the outcome of this study.

In order to achieve adequate participant enrolment to reach the target sample size (see Section 2.12.1), we will provide notice of this study among patients referred to our clinic as well as in local newspapers.

### 2.4. Participant Evaluation

All patients will undergo the following assessment measures, as reported in Table 1, performed by a psychologist, who will be also be responsible for obtaining the informed consent form.

At T0 and T1, the participants will fill in a series of psychological questionnaires as with regards their affective and emotional state:(a)Body Satisfaction Scale (BSS) [71] designed to measure satisfaction/dissatisfaction with 16 body parts;(b)Hospital Anxiety and Depression Scale (HADS) [72] for anxious and depressive symptomatology;(c)Emotive Regulation Questionnaire (ERQ) [73], which is a self-report measure of two emotion regulation strategies (i.e., cognitive reappraisal and expressive suppression);(d)Difficulties in Emotion Regulation Scale (DERS) [74] measuring emotion regulation problems.

Moreover, within each of the three sessions, they will be also evaluated before and after the visual exposure (six times) with the following measures:(a)Pain Visual Analogue Scale (VAS) for pain level on a 0 to 100 scale;(b)Body Image Questionnaire (BIQ) [75] for body image perception;(c)Positive and Negative Affect Schedule (PANAS) [76] for the positive and negative affective state.

Finally, after each visual exposure session, for a total of three times, the patients of the experimental group will fill in a questionnaire (i.e., embodiment questionnaire) related to the level of sense of belonging to the virtual body and the experience of illusion of enfacement [77]. The control group will fill in a questionnaire (i.e., immersive questionnaire) assessing the level of immersion in the virtual environment [39]. This is in order to assess the subjective strength of the two conditions to which patients will be exposed.

### 2.5. Randomization, Stratification, and Allocation

After T0, random numbers will be generated from a uniform distribution in the range 0–1, dividing the range in two equal intervals and assigning each patient to the group corresponding to the sampled number (1:1 ratio). Allocation of participants will be performed by an independent data manager, who will be not involved in the data collection/analysis.

### 2.6. Blinding

This is a double blinded study: both the patients and the clinicians collecting the data will be blinded to the group allocation. The participants do not know to what kind of VR conditions they will be exposed. On the ethics consent form, they are informed that they will be exposed in a random fashion to one of two VR conditions consisting in the exposure to a virtual environment through a head-mounted display (HMD). The clinicians coding the data are not aware of the group allocation.

### 2.7. Apparatus

We will use an HMD (Oculus Quest2 created by Facebook Technologies, a division of Meta, Facebook Inc., Menlo Park, CA, USA —see Figure 2), with a resolution of 1832 × 1920 pixels per eye resolution at 120 Hz to show the 360° video in the experimental group, which will be displayed through a VR video player app (Skybox VR player for Oculus, v. 1.0.0, Skybox Studio, Vancouver, BC, Canada), or the virtual environment displayed through the Calm Place app (v. 0.3.1) (Calm Inc., San Francisco, CA, USA) in the control group.

### 2.8. Visual-Exposure Conditions

Each VR session will be conducted by the same experimenter (i.e., a psychologist) in order to avoid any related effects. Both visual exposure conditions will be conducted with the participant seated in a chair. In case of cyber sickness, participants will be free to discontinue the VR exposure at any moment.

#### 2.8.1. Experimental Group

Patients in the experimental group will be exposed to the enfacement illusion condition, conducted similarly to other studies in other fields of research [47] (Figure 3 and Figure 4). In detail, patients will see, through the HMD, a virtual body sitting in front of them in the same position showing a happy face expression. Each session will consist of four phases:(1)Observation of the virtual facial expression and habituation to the virtual environment: the experimenter will ask patients to focus their attention on the face of the virtual body sitting in front of them for about one minute.(2)Visuo–tactile stimulation: in order to induce the enfacement illusion in the patients, the experimenter will use a brush to apply synchronous visuo–tactile stimulation to the real face of the patients, while they will be observing a synchronous tactile stimulation on the happy face displayed in the 360° video equally applied by the experimenter in the same place and at the same time. The synchronous visuo–tactile stimulation will be applied for 2 min.(3)The observation of the virtual facial expression and habituation to the virtual environment phase will be repeated to update habituation to the virtual environment.(4)The synchronous visuo–tactile stimulation phase will be repeated to update the induction of the enfacement illusion.

Overall, the patients will receive 4 min of synchronous visuo–tactile stimulation to induce the enfacement illusion to the happy face displayed in the 360° video through the HMD. The synchronous stimulation will be recorded with a metronome sound (60 beats per minute) in the 360° video. During the experimental session, the researcher will stimulate the face of the patients which at the same time will be synchronized with the beat sound displayed in the video. Patients will observe the same situation when they are immersed into the 360° virtual scenario: a researcher stimulating the virtual face placed in front of them. A beep sound will indicate the experimenter wants to start the stimulation, and he/she will follow the sound of the metronome for each stimulation. After exposure to the 360° video, the patients will fill in a questionnaire regarding the induction of the ‘enfacement illusion’ adapted from [77].

#### 2.8.2. Control Group

Participants in the control group will be exposed to a visual stimulation by observing a pleasant immersive virtual reality environment [69] (Figure 5 and Figure 6). As in the experimental group, there will be two minutes for the habituation to the virtual environment, where the experimenter will ask the patients to describe what they see. Then, the patients will observe the pleasant virtual environment for about 4 min.

After immersion into the virtual environment, the patients will fill in a questionnaire regarding the virtual reality experience adapted from [39].

### 2.9. Outcome Measures

As primary outcome measure, we will consider the effects on pain perception measured on the VAS for the treatment based on the enfacement illusion compared to the control conditions across the visual-exposure sessions.

Secondary outcome measures will include:(a)The evaluation of the effects on the perception of one’s own body image and on the affective and emotional state of the treatment based on the enfacement illusion with respect to the control conditions as evaluated with the BIQ and the PANAS at the beginning and at the end of each visual exposure condition;(b)The assessment of the virtual reality experience within each treatment condition by comparing the embodiment questionnaire (for the experimental group) and the immersive questionnaire (for the control group) across the three visual exposure sessions;(c)The evaluation of the relationship between the affective and emotional state of patients and the change in the perception of pain and one’s body image as assessed via the BSS, the HADS, the ERQ, and the DERS at T0 and T1 in the two visual exposure conditions (experimental group vs. control group).

### 2.10. Data Collection

Psychologists conducting evaluations will receive appropriate instruction and guidance regarding all of the outcome parameters and assessments that will be taken. The research staff collecting the data will be blinded to the group allocation.

### 2.11. Data Management

Study data will be recorded in a repository in an Excel file. All participants will be registered with an identification code in a random order. The repository will be kept updated to reflect the subject’s status at each stage during the course of the study. The collected data, after scientific publication, will be shared in public repositories (Zenodo) according to the good practice of data sharing.

### 2.12. Statistical Analysis

#### 2.12.1. Sample Size Calculation

According to the main goal of demonstrating possible changes in pain perception—as assessed on the VAS scale—in patients with CM after a non-pharmacological intervention, the number of participants needed is 100. This number was calculated in order to guarantee a statistical power of 80% and a statistical level of 95% for a one-tailed *t*-test in order to detect an effect size (d) = 0.5 by expecting significant differences between the two groups in terms of changes in pain perception both after each treatment session and at the end of the entire intervention in favor of patients who fall into the experimental group (delta pain for the experimental group = −40 ± 100; delta pain for the control group = 10 ± 100). With the above data, the required sample size necessary to detect significant differences is 50 subjects per group. Extra participants will be recruited in case of dropouts.

#### 2.12.2. Planned Analysis

Statistical analysis on outcome measures will be conducted using Stata 13 (StataCorp LP, College Station, TX, USA). The data collected will be first analyzed using descriptive statistical techniques. As the primary outcome, differences in terms of VAS scores at session 1 pre assessment and session 3 post assessment in the two groups will be evaluated using ANOVA-one factor (factor: groups) (or the Kruskal–Wallis test, after a normality analysis). Finally, mixed-effects models will be implemented to examine the longitudinal aspect of the study that represent changes across the visual-exposure sessions. For secondary outcomes concerning the changes in the perception of one’s own body image and on the affective and emotional state as well as the assessment of the VR experience, the analysis plan will be similar to the one planned for the primary outcome. Regarding the evaluation of the relationship between the affective and emotional state of the patients and the change in the perception of pain and one’s body image, a correlation analysis will be performed between the considered scales. For all outcome analyses, an intention-to-treat (ITT) approach will be used, according to CONSORT guidelines [78,79]. An alpha level ≤0.05 will be considered as significant.

### 2.13. Ethical Issues and Dissemination Plan

This trial will involve human participants, VR-based systems, data collection, elaboration, and abstraction used for the evaluation of the two VR options. This study has been approved by the local ethics committee (IRCCS San Matteo Hospital, Pavia) and will be conducted in accordance with the Declaration of Helsinki and reported according to CONSORT guidelines [78,79]. Any subsequent modification to the protocol which may impact the study (e.g., objectives, study population, sample sizes, procedures) will be submitted as an amendment and reviewed by the IRCCS San Matteo Hospital ethics committee for approval. In addition to ethical approval, all the procedures and the data managed have been approved by the data protection officer of the IRCCS Mondino Foundation who guarantees the study’s compliance to the GDPR (General Data Protection Regulation). The information provided when presenting the informed consent form to the participants will be given in a language appropriate to the individuals’ level of understanding. In particular, in the consent form, participants are informed that they will receive, in a random fashion, one of two VR conditions consisting of exposure to a virtual environment through an HMD. We do not provide additional information about the content of each VR exposure in order to avoid influencing the subjective strength of the virtual reality experience, as assessed by the embodiment and immersive questionnaires. Participants will be encouraged to ask questions before signing the informed consent form and they will be free to discontinue the study at any moment. In order to improve the adherence to the protocol, the experimenter responsible for collecting the informed consent forms will highlight to the patients the opportunity to take part in this study as a way to improve the knowledge of their disease as well as to provide complementary non-pharmacological approaches in the future. Due to the low risk associated with study participation, an annual audit of the study, the Data Monitoring Committee (DMC), and interim analyses are not required. Additionally, a data safety monitoring board is not required because we are not investigating a medical product and we do not foresee any major risks associated with study participation.

To protect the privacy of patients, we will use a unique research code for each participant on all research-related documents. This enables us to identify individuals without using their names. The list linking the participant code to personal information will be kept in a secure electronic database with access limited. All electronic research-related participant information will be stored on a protected network in a secure file with limited access.

After the data analysis phase, patient privacy will be further preserved. The results obtained from the study will be described on a group level to prevent the data being traced back to a single person. Once the final report of the study is available, the results will be disseminated in the scientific community through publication in open-access, peer-reviewed scientific journals and presentations at national and international conferences.

Datasets generated and/or analyzed during the current study will be anonymized and stored on an online repository (Zenodo) according to the good practice of data sharing. External researchers may obtain access to the final trial dataset. The (intellectual) property rights with regard to the generated data will reside at the IRCCS Mondino Foundation.

To the best of our knowledge, VR-based interventions should not have any potential negative impacts on the participants. A possible adverse effect related to VR could be cyber sickness (nausea and other side effects) [80]. In order to avoid this, VR exposure will be limited to 20 min at a time and patients will be seated while using the HMD. The investigator will communicate any possible, unforeseen, adverse event to the Ministry of Health.

Regarding payment policies for participants, the compensation’s amount and the method and timing of disbursement must be consistent with the laws, regulations, and guidelines of the region in which the study is conducted and must not improperly influence the decision to participate. This trial is a no-profit study, and, in Italy, the national legislation states that it is forbidden to offer or request any kind of financial benefit for participation in a clinical, experimental trial. The costs associated with the implementation of this RCT will be supported by the IRCCS Mondino Foundation.

## 3. Discussion and Conclusions

CM represents a life-altering condition resulting in a significant reduction in the quality of life [81]. Sufferers are usually treated with acute medications and preventive treatments, which can be poorly tolerated and present significant risks and side effects [4,5,82]. Hence, CM patients are left with few efficacious and safe therapeutic options. To this end, the technological advancements of the last few years are providing promising and novel non-pharmacological options to be adopted for treating clinical conditions, including pain states. VR systems aimed at reducing virtual BOIs can then represent new tools for the study of perceptual processes acting on body ownership [17,26] and emotional states [52,53]. In this field, synchronous multisensory stimulations, such as visuo–tactile stimulation, can be used temporarily to induce enfacement, which is the subjective illusion of ownership of another person’s face [46,47]. The experimental induction of the enfacement illusion can represent an interesting application strategy for pain relief in CM.

The present RCT will provide evidence about the use of the enfacement illusion in virtual reality for pain relief in CM. It is expected that such results will arise because of the improvement in body image perception and empathy for positive emotions. Results of this RCT will advance the knowledge of this painful condition by exploring the relationship existing between internal body representation, distortion of the body image, and pain perception in CM. Furthermore, the results from this study will support the effectiveness of body ownership illusions, such as the use of the enfacement illusion, as a cognitive behavioral intervention for pain relief.

Chronic migraine’s impact on society has ever been overcast by stigma, its historically obscure etiology, and the comorbidities often associated with the disease. From an economic point of view, migraine is considered as one of the costliest neurological diseases in Europe [83,84]. It is indeed characterized by diminished productivity and increased utilization of healthcare resources, translating into higher costs for both the individual and society [83,85,86]. To the best of our knowledge, this is the first study evaluating the effects of a treatment based on the enfacement illusion in reducing pain perception in this kind of patient. It is well known that CM represents a very challenging condition from a therapeutic point of view. Evidence suggests the usefulness of a multidisciplinary approach [87] in the management of difficult-to-treat primary headaches. Findings from this RCT could pave the way to the definition and testing of non-pharmacological therapies mediated by virtual BOIs in the management of pain in CM. Given the promise of virtual embodiment and enfacement illusions in reducing pain perception by means of body representation, this study will determine the extent to which VR interventions can positively affect pain perception. Furthermore, such VR interventions may alleviate this condition and provide additional alternate mechanisms for providing benefits to CM patients.

This study is a first, important step toward the evaluation of the effectiveness of the proposed virtual BOI for pain relief in CM. Larger clinical trials will be needed to confirm the findings and more behavioral studies, as well as a better understanding of this disorder, and more tailored, individual-based approaches to its treatment will be needed in the future. In this framework, the implementation of VR systems in the management of CM could be incorporated into clinical routine practices as a complementary, non-pharmacological therapy to reduce pain perception, providing promising outcomes for the patients.

### Strengths and Limitations

In the field of CM, the present randomized controlled trial will allow the implementation and assessment of the effectiveness of an enfacement illusion intervention targeting pain in this condition. The availability of alternative non-pharmacological treatments will represent an expanding clinical practice and an interesting area of research. Whenever confronted with headache patients with a complex disease or those who are unwilling to accept possible drug-induced adverse events, clinicians should consider this rapidly growing armamentarium of alternative strategies of treatment and choose treatment methods based on desired the clinical indications and patient’s characteristics.

This study has some limitations that need to be acknowledged. First, there are no plans to collect any functional measurements of activity changes in the pain modulatory system, which could shed light on the neural phenomena associated with the enfacement illusion. Second, this study involves only female CM participants. Even if it is known that there are differences in pain perception between males and females [70], we are aware that our choice will limit the translatability of the findings. We also decided to exclude pregnant or breastfeeding women in order to avoid the possibility of secondary headache [88] in which the effects of our intervention would be distorted by confounding variables. However, such a decision should limit the generalizability of our findings. Third, the intervention lasts for only three sessions and our primary outcome should not be considered representative for an efficacy study, according to control trial guidelines [89]. Hence, it is expected that they study’s effects could be limited from a temporal point of view. Finally, we are also aware that we are not using a validated pain measure, such as the McGill pain questionnaire [90], for assessing our primary outcome. However, the VAS has been used in a large number of studies to rate the level of pain perception immediately after exposure of the experimental stimuli [91,92,93]. We consider that fact that as the participants have to rate their level of pain immediately after exposure to the VR conditions, an intuitive VAS assessment will be the more appropriate measure. Based on the results from this pilot clinical study, it will be possible to design further RCTs with longer intervention periods and to even investigate the possible neural correlation behind such effects as well as to provide a broad view on how the enfacement illusion could be effective in inducing benefits for migraine frequency and also for the non-painful, non-cephalic aspects of migraine (e.g., nausea, photophobia, etc.). Nonetheless, we felt that, from a research point of view and as a first step in this completely new field of research, it was important to start with a homogeneous population to control for the variables.

## Figures and Tables

**Figure 1 jcm-11-06876-f001:**
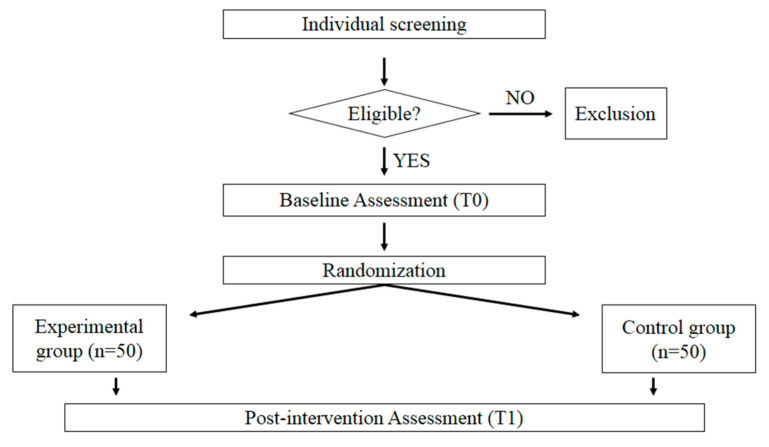
CONSORT flow chart for enrollment and randomization.

**Figure 2 jcm-11-06876-f002:**
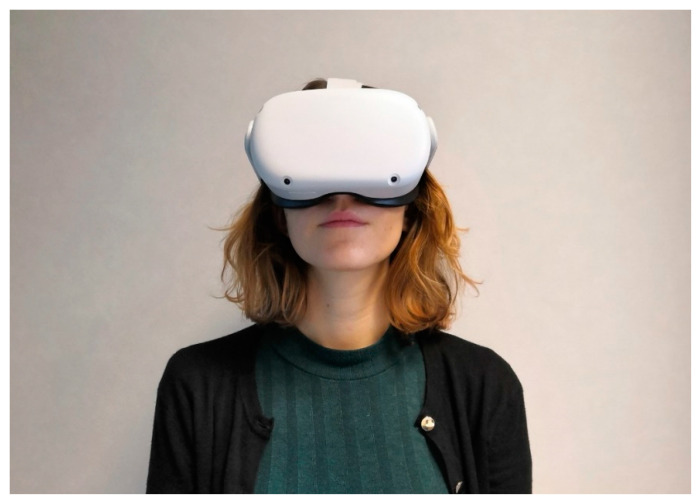
The HMD to be used in the experimental group and the control group.

**Figure 3 jcm-11-06876-f003:**
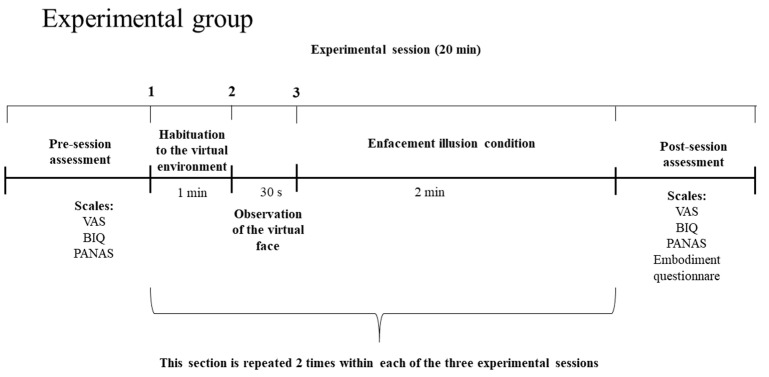
Timeline of each experimental session for the experimental group.

**Figure 4 jcm-11-06876-f004:**
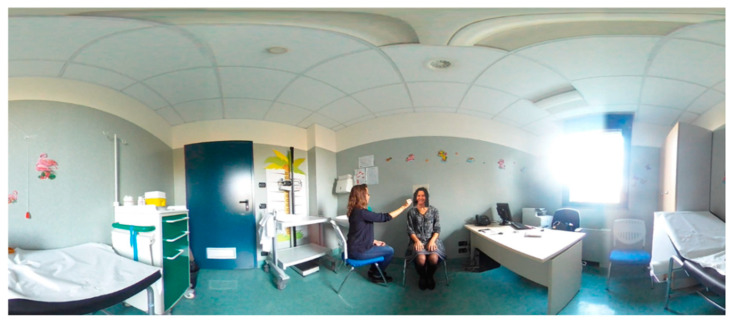
Caption of the 360° video while applying the face visuo–tactile stimulation to induce the enfacement illusion.

**Figure 5 jcm-11-06876-f005:**
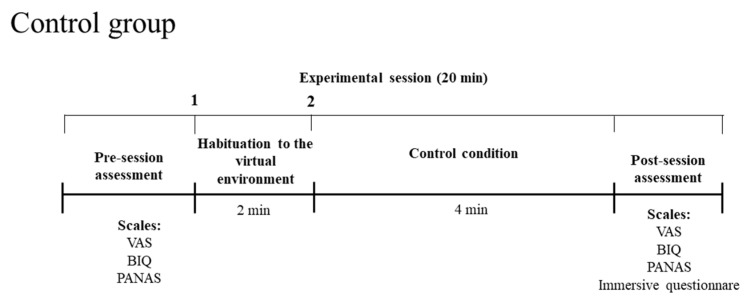
Timeline of each experimental session for the control group.

**Figure 6 jcm-11-06876-f006:**
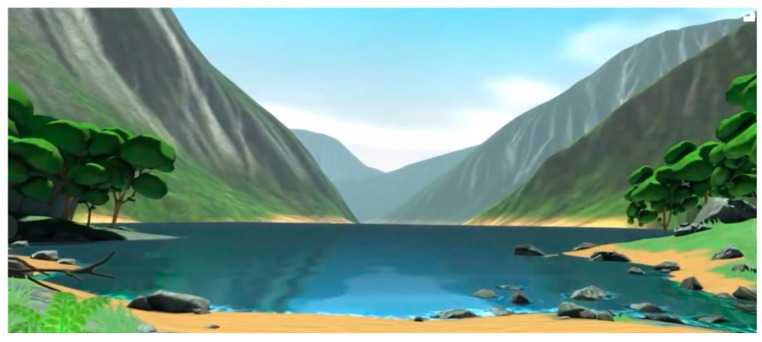
Caption of the pleasant virtual environment retrieved from the Calm Place app.

**Table 1 jcm-11-06876-t001:** Materials for participant evaluation as a function of the testing session.

		Session 1		Session 2		Session 3		
Outcome Measures	T0	Pre	Post	Pre	Post	Pre	Post	T1
Primary outcome								
Pain Visual Analogue Scale (VAS)		x	x	x	x	x	x	
Secondary outcome								
Hospital Anxiety and Depression Scale (HADS)	x							x
Emotive Regulation Questionnaire (ERQ)	x							x
Difficulties in Emotion Regulation Scale (DERS	x							x
Body Satisfaction Scale (BSS)	x							x
Body Image Questionnaire (BIQ)		x	x	x	x	x	x	
Positive and Negative Affect Schedule (PANAS)		x	x	x	x	x	x	
Embodiment/Immersive questionnaire			x		x		x	

Note: the embodiment questionnaire is administered to the experimental group, whereas the immersive questionnaire is administered to the control group.

## Data Availability

Datasets generated and/or analyzed during the current study will be anonymized and stored on an online repository (Zenodo, https://zenodo.org/), according to the good practice of data sharing. External researchers may obtain access to the final trial dataset. The (intellectual) property rights with regard to the generated data will reside at the IRCCS Mondino Foundation. Anonymized results will be published in peer-reviewed journals and presented at international conferences.

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
