# Peer review of "The Virtual “Enfacement Illusion” on Pain Perception in Patients Suffering from Chronic Migraine: A Study Protocol for a Randomized Controlled Trial"

_jcm, 2022, doi:10.3390/jcm11226876_

Round 1

Reviewer 1 Report

This protocol describes a randomized, double-blind controlled trial of a novel virtual “Enfacement Illusion” on pain perception in patients with CM. It provided sufficient literature review in the background and addressed the scientific rationale behind this virtual therapy. The study will recruit only female patients with CM where 3 sessions of “enfacement illusion” vs. “pleasant environment” will be compared for immediate pain perception change as their primary outcome. Sample size calculation, statistical analysis, and ethical issues were properly described. If successful, this will be the first to demonstrate the potential utility of virtual body ownership illusion for pain relief in CM. Overall, this is a well-written protocol from an established headache research group.

Specific comments:

1.     The primary outcome is not a typical outcome for acute migraine treatment recommended by the control trial guideline. It may not be long enough for an efficacy study, which is usually 2 hours post-treatment, but acceptable for a simple pain perception assessment.

2.     It is not clear how 3 treatment sessions will be offered. It appears to be offered consecutively with VAS assessment after each session. If that is the case, there could be an additive effect on pain perception after each session. It might be worth a discussion.

3.     The experiment group is a more elaborated visual-exposure condition, while the control group is only a pleasant virtual environment. If the key question is to determine if body ownership illusion (synergism of happy face and embodiment) can alter pain perception, would it be better if the control group were a smiley face without tactile stimulation (or tactile stimulation without a smiley face)?

4.     The CM is a very diverse patient population. Patients with daily headaches, with another headache diagnosis (e.g., post-traumatic headache, medication overuse headache), with noncephalic pain disorder, or refractory to multiple preventive classes may not be the same as patients with low chronic frequency. Would it be a more homogenous study population if certain specific headache conditions were excluded?

5.     The sample size calculation, based on an effect size of 0.5 for 2 independent groups on t-test, seems to be generated from a one-tail test. If calculated with a two-tail test, the sample size is calculated to be 64 subjects per group. Could the author please clarify the sample calculation a bit more?

6.     Table 1: BIQ is done or not done at T1?

7.     Figure 3 “the section is repeated 2 times….” Is it 2 times or 3 times during the experiment? Also, what is the enfacement illusion questionnaire? Is it the embodiment questionnaire? Please clarify.

8.     2.9 data management. “access” database. Is it Microsoft Access?

9.     Line 312: “Kurskal Wallis” should be Kruskal-Wallis.

10.  Line 385: “this this” should be only “this”.

11.  Ref. #68. The full reference is available. Please provide updated information. Matamala-Gomez M, Bottiroli S, Sances G, Allena M, De Icco R, Ghiotto N, Guaschino E, Sandrini G, Tassorelli C. Facial expressions modulate pain perception in patients with chronic migraine. Cephalalgia. 2022 Jul;42(8):739-748. doi: 10.1177/03331024221075081.

Author Response

My co-authors (Matamala-Gomez, Allena, Guaschino, Ghiotto, De Icco, Sances, and Tassorelli) and I have now completed our revision of ms “The virtual "Enfacement Illusion" on pain perception in patients suffering from chronic migraine: A study protocol for a randomized controlled trial” to the Journal of Clinical Medicine. 

We were gratified by the enthusiasm you and the reviewers gave for our work.  We certainly, and we definitely appreciate a chance to resubmit the manuscript, especially given that we feel our approach is truly novel, so it was great getting a chance to clarify its novelty and unique contribution to VR approaches aimed at reducing pain perception in chronic migraine. 

We have done our best to act on every one of the Reviewers’ requests, and to help you evaluate our revision, we provide a point-by-point discussion (below in italics) of how we addressed each one. 

We hope you agree that the revised manuscript is markedly improved, and we look forward to hearing back from you about any other comments you may have. 

Thanks much for working with us!

Sara Bottiroli

Reviewer 1

This protocol describes a randomized, double-blind controlled trial of a novel virtual “Enfacement Illusion” on pain perception in patients with CM. It provided sufficient literature review in the background and addressed the scientific rationale behind this virtual therapy. The study will recruit only female patients with CM where 3 sessions of “enfacement illusion” vs. “pleasant environment” will be compared for immediate pain perception change as their primary outcome. Sample size calculation, statistical analysis, and ethical issues were properly described. If successful, this will be the first to demonstrate the potential utility of virtual body ownership illusion for pain relief in CM. Overall, this is a well-written protocol from an established headache research group.

 We would like to thank the reviewer for the appreciation of our work. We believe that VR could represent an interesting and prosing field of research for providing novel intervention aimed to reduce pain in chronic conditions, such as chronic migraine.

Specific comments:

  1. R: The primary outcome is not a typical outcome for acute migraine treatment recommended by the control trial guideline. It may not be long enough for an efficacy study, which is usually 2 hours post-treatment, but acceptable for a simple pain perception assessment.

A: We thank the Reviewer for this consideration.

We agree that our primary outcome should not be considered representative for an efficacy study, according to control trial guidelines (Diener et al., 2019) that refer to be pain free at 2 hours after treatment. Hence, it could be considered as a pain perception assessment, as suggested by the Reviewer.

However, it should be noted that this is a first study using the ‘enfacement illusion’ for pain relief in patients with chronic migraine. Hence, we expect that our results will last for a while after the exposure, but, in order to induce longer pain relief responses, we should design a more intensive and longer treatment. For this reason, consistently in the paper, we referred to the effects of this intervention in terms of changes in pain perception.  In any case, from a research point of view, we like to think that this is a first step in a completely new field of research in CM. We have further corroborated this consideration among study limitations (see pag 14).

  1. R: It is not clear how 3 treatment sessions will be offered. It appears to be offered consecutively with VAS assessment after each session. If that is the case, there could be an additive effect on pain perception after each session. It might be worth a discussion.

A: We agree with the Reviewer that this pint needed to be clarified.

The three VR sessions will be offered during a one-week period. The first VR session will be carried out immediately after enrolment (in which the neurologist will assess the characteristics of the attack that must be of migraine nature, as reported among inclusion criteria) and baseline (T0) evaluation. The next two VR sessions will be carried out in the following days when the patient will report to have a migraine attack. It is easy to expect them to have an attack for two main reasons: (i) at the baseline the neurologist will assess that they are having a migraine-type attack; (ii) they are CM patients and not allowed to take abortive medications during the study period.

Given that VR sessions will be conducted at least 24 hours apart, we do not expect additive effects on pain perception, as resulted from the VAS assessment, due to the previous session. Also on this point, it  should be noted that collected data will be analysed both considering the longitudinal trend of pain scores across the three sessions, but also within each session comparing pre and post between groups. In this way is could be possible to appreciate the effects of the VR exposure within each single session.

We have added more information about this part also in the text so that it results clearer (see pag.4).

  1. R: The experiment group is a more elaborated visual-exposure condition, while the control group is only a pleasant virtual environment. If the key question is to determine if body ownership illusion (synergism of happy face and embodiment) can alter pain perception, would it be better if the control group were a smiley face without tactile stimulation (or tactile stimulation without a smiley face)?

A: Thanks for this consideration.

We did not include a control group based on smiley face without tactile stimulation or tactile stimulation without a smiley face in light of these two considerations. First, the exposure to pleasant stimuli seems to be able to produce positive effects on pain perception in CM as resulted in previous investigations (de Tommaso et al., 2009). Second, this same positive effect has been shown also by exposing patients to facial expressions, as we found in a work we recently published (Matamala-Gomez et al., 2022).

Then, as we already investigated the effect of observing a positive facial expression for pain relief in patients with CM (Matamala-Gomez et al., 2022), in the present study we aim to compare that stimulus (that we already know works for pain relief) with another positive and pleasant visual stimulus that can be developed in immersive virtual reality too. In our opinion, the tactile stimulation alone won’t be a good control condition. In this case, it will be used in order to produce the enfacement illusion and we do not believe that being touched with a brush synchronized with a metronome for 4 minutes could reduce pain per se. As, for sure, the positive visual exposure (pleasant VR environment) is expected to induce a higher pain relief effect than the touch stimulation alone.

We have better explained the choice of this control condition at the end of the introduction (pag. 3): The rationale for choosing this control condition derives from the willingness to use another positive visual stimulus in immersive VR, already known to be effective in distracting from pain in CM (de Tommaso et al., 2009), that can serve to appreciate the additional effects of producing the enfacement illusion of a happy face in the experimental group.

  1. R: The CM is a very diverse patient population. Patients with daily headaches, with another headache diagnosis (e.g., post-traumatic headache, medication overuse headache), with noncephalic pain disorder, or refractory to multiple preventive classes may not be the same as patients with low chronic frequency. Would it be a more homogenous study population if certain specific headache conditions were excluded?

A: We thank the Referee for this suggestion.

In the present study we aim to include only patients affected by CM with or without medication overuse headache (MOH). Other primary and/or secondary headache diagnoses according to ICHD-3 criteria will represent an exclusion criterion. We decided to include also patients with MOH to preserve study feasibility (we know that a great number of patients with CM managed every year at our third-level headache clinic also suffers from MOH).

From a clinical perspective, we agree that different phenotypes may exist across the CM spectrum (namely patients with daily headache, patients with multiple preventives failures, and so on), but these subtypes are not defined in ICHD-3. Thus, we decided to be conservative on this topic and include patients with a CM diagnosis according to ICHD-3 without further classification.

By contrast, we agree that patients with non-cephalic pain (such as fibromyalgia, chronic low-back pain, and so on) may represents a completely different study population. For this reason, we welcome the Referee’s suggestion and we now include the presence of chronic non-cephalic pain as an exclusion criterion.

  1. R: The sample size calculation, based on an effect size of 0.5 for 2 independent groups on t-test, seems to be generated from a one-tail test. If calculated with a two-tail test, the sample size is calculated to be 64 subjects per group. Could the author please clarify the sample calculation a bit more?

A: Sorry for this misunderstanding.

Actually, given that we postulated specific hypothesis about the direction of the differences between groups, we used a one-tail test.

We have now re-formulated this part into (see pag. 10): “This number was calculated in order to guarantee a statistical power of 80% and a statistical level of 95% for a one tailed t test in order to detect an effect size (d) = 0.5 by expecting significant differences between the two groups in terms of changes in pain perception both after each treatment session and at the end of the entire intervention in favour of patients who fall into the experimental group (delta pain for the experimental group = -40 ±100; delta pain for the control group = 10±100)”.  

  1. R: Table 1: BIQ is done or not done at T1?

A: Sorry for this typo.

BIQ is carried out at the beginning and at the end of each experimental session. We have fixed this inconsistence in Table 1.

  1. R: Figure 3 “the section is repeated 2 times….” Is it 2 times or 3 times during the experiment? Also, what is the enfacement illusion questionnaire? Is it the embodiment questionnaire? Please clarify.

A: Thanks for this comment because this part could be not very clear.

We specified in the Method section that the procedure for the experimental group reported in Figure 3 is repeated 2 times within each experimental session. So that it is carried out 2 times for each of the three sessions.

We have also fixed Figure 3 in which the parenthesis was unclear and included also the evaluation pre-post phase.

We apologize for this typo. The enfacement illusion questionnaire refers to the embodiment questionnaire. We changed the inconsistence in Figure 3 using the same nomenclature (i.e., embodiment questionnaire).

  1. R: 2.9 data management. “access” database. Is it Microsoft Access?

A: We clarified it is referred to MS Excel (see pag. 10).

We changed the sentence into: “Study data will be recorded in a repository consisting in an Excel file”. We apologize because we used the term “access” incorrectly.

  1. R: Line 312: “Kurskal Wallis” should be Kruskal-Wallis.

A: This typo has been fixed.

  1. R: Line 385: “this this” should be only “this”.

A: We apologize. We deleted the second “this” accordingly.

  1. R: Ref. #68. The full reference is available. Please provide updated information. Matamala-Gomez M, Bottiroli S, Sances G, Allena M, De Icco R, Ghiotto N, Guaschino E, Sandrini G, Tassorelli C. Facial expressions modulate pain perception in patients with chronic migraine. Cephalalgia. 2022 Jul;42(8):739-748. doi: 10.1177/03331024221075081.

A: Thanks for having noticed this aspect. This reference has been updated.

Reviewer 2 Report

Thank you for sending this protocol for review. I found it an interesting and novel area of research. Below are my comments:

The authors should consider the use of the SPIRIT reporting guideline as this would provide a better structure for the protocol and ensure that all necessary information is reported e.g., monitoring and management of adverse events, dissemination plans.

The main outcome is a pain VAS. Is there a reason the authors did not use a validated pain measure? As this is the main outcome, one would have thought this was important and would help in the publication of the future trial. Is there a scale particularly for migraines and the nuances involved in this condition? As it currently is reported, the authors need to include more detail about their primary outcome. 

Throughout there are a few grammatical issues (missing words).

Table 1 suggests the primary outcome will not be collected as part of the baseline (T0) or post-sessions (T1)? How will the authors control for baseline pain?

Author Response

My co-authors (Matamala-Gomez, Allena, Guaschino, Ghiotto, De Icco, Sances, Tassorelli ) and I have now completed our revision of ms “The virtual "Enfacement Illusion" on pain perception in patients suffering from chronic migraine: A study protocol for a randomized controlled trial” to the Journal of Clinical Medicine. 

We were gratified by the enthusiasm you and the reviewers gave for our work.  We certainly, and we definitely appreciate a chance to resubmit the manuscript, especially given that we feel our approach is truly novel, so it was great getting a chance to clarify its novelty and unique contribution to VR approaches aimed at reducing pain perception in chronic migraine. 

We have done our best to act on every one of the Reviewers’ requests, and to help you evaluate our revision, we provide a point-by-point discussion (below in italics) of how we addressed each one. 

We hope you agree that the revised manuscript is markedly improved, and we look forward to hearing back from you about any other comments you may have. 

Thanks much for working with us!

Sara Bottiroli

Reviewer 2

Thank you for sending this protocol for review. I found it an interesting and novel area of research.

We thank the reviewer for these considerations.

Below are my comments:

R: The authors should consider the use of the SPIRIT reporting guideline as this would provide a better structure for the protocol and ensure that all necessary information is reported e.g., monitoring and management of adverse events, dissemination plans.

A: We thank the reviewer for this important suggestion.

We have fully revised the Method according to the SPIRIT guidelines, as suggested, as well as the SPIRIT checklist has been added as supplementary material. This has been useful in order to add all missing information.

Just to cite a few, we have added the dissemination plan, strategies for recruitment, possible side effects and more specifications about data collection and management).

 R: The main outcome is a pain VAS. Is there a reason the authors did not use a validated pain measure? As this is the main outcome, one would have thought this was important and would help in the publication of the future trial. Is there a scale particularly for migraines and the nuances involved in this condition? As it currently is reported, the authors need to include more detail about their primary outcome. 

A: We are aware that we decided to adopt as primary outcome the VAS instead of a validated pain measure (e.g., McGill Pain Questionnaire that allows to collect more aspects related to the dimension of pain). However, as we explained in the text among study limitations, the pain VAS has been used in a large amount of studies to rate the level of pain perception suddenly after the exposure of the experimental stimuli. We consider that as participants have to rate their level of pain immediately after the exposure of the visual stimulus, an intuitive VAS assessment will be the more appropriate measure.

Definitely, deeper assessment of the level of pain will be useful for a longer control trial study in order to better characterize the level of pain or changes on pain perception. We have added these considerations among study limitations (see pag. 13).

R: Throughout there are a few grammatical issues (missing words).

A: Sorry for this. We have full revised the paper for typos.

R: Table 1 suggests the primary outcome will not be collected as part of the baseline (T0) or post-sessions (T1)? How will the authors control for baseline pain?

A: We clarified this aspect in the text (see pag. 4). Session 1 will be carried out immediately after enrolment and then T0, as well as T1 will follow Session 3.

Patients after enrolment and randomization will go through T0 and then they will receive the first VR session with corresponding assessments (pre and post), including their level of pain. At the end of the third VR session, and then of Session 3 post assessment, patients will go through T1. According to this, migraine pain level will be assessed at the beginning of each session, besides being an inclusion criteria at enrolment. As consequence, we do not believe that baseline pain could have an impact across sessions.

Furthermore, it is important to note that, as reported in the Planned analysis section, data will be analysed both analysing the longitudinal trend of pain scores across sessions, but also within each session. So that we do not believe that we would be able to appreciate the effects of the VR condition independently from baseline levels of pain.

Round 2

Reviewer 1 Report

The authors have carefully addressed all comments.

Author Response

We would like to thank the reviewer for approving the final version of our paper.

Best regards,

Sara Bottiroli